# A New Scenario-Based Approach for Water Quality and Environmental Impact Assessment Due to Mining Activities

Mohd Talha Anees [1] , Mohammad Muqtada Ali Khan [1,*] , Mohd Omar Abdul Kadir [2], Kamal Abdelrahman [3] , Ahmed M. Eldosouky [4] , Peter Andráš [5], Nasehir Khan Bin E. M. Yahaya [6], Zubaidi Johar [7], Mohammed S. Fnais [3] and Fatehah Mohd Omar [8]

1 Faculty of Earth Science, Universiti Malaysia Kelantan, Jeli Campus, Locked Bag No. 100, Jeli 17600, Malaysia; talhaanees_alg@yahoo.in
2 School of Industrial Technology, Universiti Sains Malaysia, Minden 11800, Malaysia; pultexsb@yahoo.com
3 Department of Geology & Geophysics, College of Science, King Saud University, P.O. Box 2455, Riyadh 11451, Saudi Arabia; khassanein@ksu.edu.sa (K.A.); mfnais@ksu.edu.sa (M.S.F.)
4 Geology Department, Faculty of Science, Suez University, Suez 43518, Egypt; dr_a.eldosoky@yahoo.com
5 Faculty of Natural Sciences, Matej Bel University in Banska Bystrica, Tajovského 40, 974 01 Banska Bystrica, Slovakia; peter.andras@umb.sk
6 National Hydraulic Research Institute of Malaysia (NAHRIM), Ministry of Environment and Water, Lot 5377, Jalan Putra Permai, Seri Kembangan 43300, Malaysia; nasehir@nahrim.gov.my
7 National Hydraulic Research Institute of Malaysia (NAHRIM), Ministry of Natural Resources & Environment (NRE), Seri Kembangan 43300, Malaysia; zubaidi@nahrim.gov.my
8 School of Civil Engineering, Universiti Sains Malaysia, Nibong Tebal 14300, Malaysia; cefatehah@usm.my
* Correspondence: muqtadakhan@gmail.com

**Abstract:** Water quality assessment and its monitoring are necessary for areas of mining activities. In Malaysia, the mining industry is the backbone of the manufacturing and construction sectors. This study used spatio-temporal water quality modeling along a reach with mining activities during high and low discharges at Sungai (river) Lebir and Sungai Aring, situated in Gua Musang, Kelantan, Peninsular Malaysia. The objective was to assess the spatio-temporal environmental impact of mining activities during the wet and dry seasons. Data were collected at different locations along the reach. Point and non-point sources were near the mining site. Overland flow calculation at the mining site was found with the widely used SCS (Soil Conservation Service) curve number method. Several scenarios were analyzed, such as baseline, worst-case, and with-mitigation. The study revealed that baseline values of all parameters were either in a natural condition or slightly polluted, except for aluminum. All parameters were estimated at a high concentration from the mining site to downstream during the worst case of the wet season. Whereas, during the worst case of the dry season, no significant differences were observed compared to baseline values. In the with-mitigation scenario, parameter concentrations were improved and similar to baseline values. Overall, the scenario selection was helpful in the environmental impact assessment. Furthermore, this study will be significant in pre- and post-mining assessment and environmental clearance.

**Keywords:** environmental impact assessment; heavy metal; mining; pollution; water quality

## 1. Introduction

River water quality is crucial for all human activities such as drinking water supply, agriculture, and industrial usage. However, the upstream fresh river water is being polluted due to point and non-point sources downstream. However, these sources cannot be permanently closed due to economic development. Every country tries to use its available resources to fulfill its requirements. To control the downstream pollution sources, spatio-temporal monitoring of point and non-point sources is mandatory in different conditions

or scenarios. Every region has diverse climatic conditions that require a particular environmental impact assessment technique to control pollution sources and plan for proper mitigation measures.

The mining industry plays a crucial role in economic development and provides millions of jobs. In Malaysia, the mining industry is one of the oldest industries, which provides the backbone for the manufacturing and construction sectors. According to the Mineral and Geoscience Department (JMG), an estimated reserve of the country's mineral resources is 4.11 trillion [1]. Several mines are operational for metallic, non-metallic, and energy minerals. Mining operations serve the country's economic advancement. However, it also causes surface water and soil pollution due to the point and non-point sources. Due to the tropical climate of Malaysia, high precipitation causes high surface runoff that is vulnerable to surface water pollution. According to the Department of Environment (DOE), 186 and 43 rivers are slightly polluted and polluted respectively in Peninsular Malaysia, which is approximately a 43% increase in 3 years [2]. Due to the increase in anthropogenic activities, surface water quality requires frequent monitoring.

Surface mining activities release both point and non-point sources, which have been analyzed by several studies [3–9]. However, non-point source modeling has been a major challenge for researchers and administrations due to complex hydrological processes and land use and land cover patterns [10]. Additionally, high surface runoff accelerates pollutant concentrations such as sediment, nitrogen, and phosphorous from agricultural land, and heavy metals, rubber fragments, sodium, and sulfate from urban land [3]. Heavy metals also pollute soil through several geogenic processes. It can cause hazardous enrichment into surface/sub-surface waters. The enrichment can also be found in remote areas far from the anthropogenic source [11,12]. Moreover, water chemistry can be affected through the hydrogeological path in crystalline aquifers [13]. Furthermore, accurate spatio-temporal estimation of overland flow discharge is a challenging task and variable in different scenarios.

The selection of different scenarios to model high and low flow is also important. High pollution load causes more toxic waste discharge into the river during high precipitation and overland flows. Scenarios were selected according to the study's objectives. For instance, in a high rainfall period, the different flow rate is used to estimate pollution load [14]. Chowdhury et al. [15] considered the positioning of point sources as a scenario. A hypothetical climate change scenario has been used based on air temperature and streamflow in water quality analysis [16]. Changes in land use and land cover scenarios were also used in water quality analysis. For example, during water quality analysis of landfill leachate, landfill waste composition, operation mode, and landfill age are additional scenarios [17]. Other parameters include soil erosion, industrial wastewater, animal feedlot, and rural household sewage [18].

These scenarios are generally analyzed either through water quality modeling or indices. The Hydrologic Engineering Centers River Analysis System (HEC-RAS) and Qual2k are used for water quality modeling of rivers and streams. Both models calculate fate and contaminants with different equations. Most of the studies were conducted to find the pollution load and water quality pollution due to industrial and landfill waste. However, to the author's knowledge, limited work was found on spatio-temporal water quality modeling along a reach due to mining activities during high and low discharges. Several mining projects, such as mining iron, manganese, gold, tin, and granite, are operating near different rivers in Malaysia. Additionally, many water treatment plants are situated at several rivers, which need to be monitored.

Based on the research gap, questions may arise, such as (i) how much variation is there in spatio-temporal pollution load during the wet and dry season and (ii) what is the effect of pollution on water treatment plants in different discharges? Therefore, the objectives of this study are (i) to conduct water quality modeling along the reaches using Qual2k, (ii) to examine the spatio-temporal variation of parameter concentration during the wet and dry season, and (iii) to analyze the pollution effect on water treatment plants during the

wet and dry season. The approach of this study will be helpful for mine owners, planners, and decision-makers to maintain water quality along the affected reaches and near water treatment plants.

## 2. Materials and Methods

### 2.1. Study Area

The study area lies between the latitudes 4°57′46.4″ and 5°29′13.0″ North and longitudes 102°6′20.4″ and 102°25′34.5″ East of district Gua Musang, Kelantan, Peninsular Malaysia. The elevation ranges from 4 m to 2095 m, and the slope varies between 1 and 33 degrees (Figure 1).

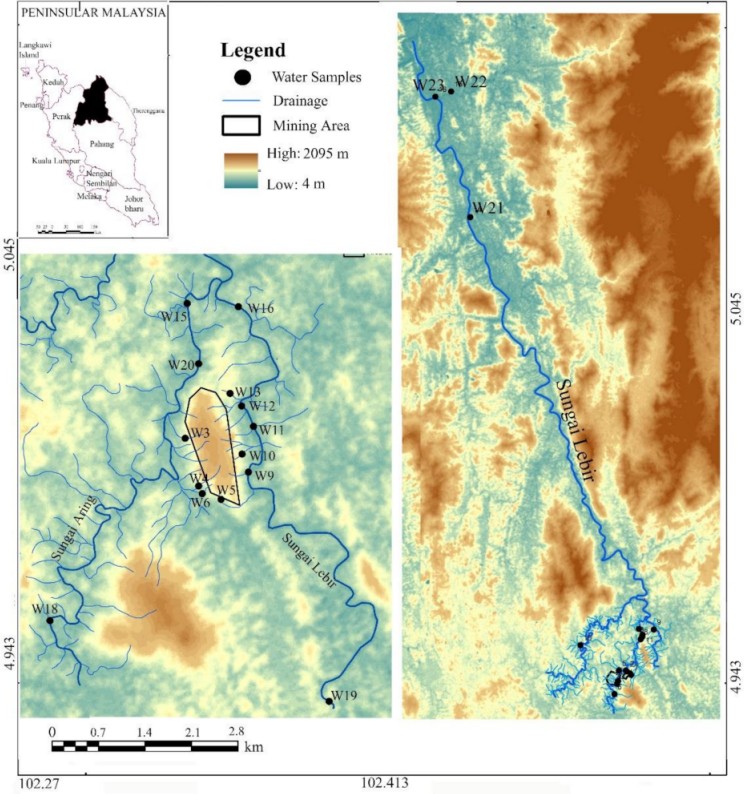

**Figure 1.** Location of water sampled and study area.

The climate is tropical and humid, with an average temperature ranging from 20 °C to 30 °C. The period from November to January receives maximum rainfall, while June and July are the driest months [19]. The average annual rainfall of the area is 3017.84 mm, while the average daily annual wind speed is 1.50 m/s. The main river in the study area is Sungai Lebir, in which two streams are joined, the Sungai Aring and Sungai Relai. Based on the locations of water treatment plants and the location of the mining area, the selected lengths of Sungai Lebir and Sungai Aring were 89 km and 14.5 km, respectively. The mining area is situated between Sungai Lebir and Sungai Aring (Figure 1). Two water treatment plants are located at Sungai Lebir, the WTP Manik Urai (68 km from upstream) and WTP Pahi (81 km from upstream), while one water treatment plant is located at Sungai Aring (13 km from upstream). Sungai Relai was not included in the modeling because it is far from the mining site and joins the Sungai Lebir after the junction of Sungai Lebir and Sungai Aring. The proposed manganese ore mining area was 202.37 hectares, located on a hill. The site is surrounded by secondary forests, rubber plantations, and oil palm plantations. Several tributaries from the mining area join both Sungai Lebir and Sungai Aring (Figure 1).

Geologically, the area lies under the Aring Formation (total thickness is 3000 m), with age ranging from late Carboniferous to early Triassic. Late Carboniferous rocks mainly

consist of volcanic and argillaceous rocks with some calcareous and arenaceous sediment from shallow marine environments [20]. The top portion of the formation is about 1000 m which is interbedded with tuffaceous limestone/slate/limestone. It is unconformably overlain by the Telong Formation [21].

### 2.2. Data Collection and Processing

Before sample collection, a preliminary survey was performed to confirm the locations. A total of 23 water samples were collected at the mining site and along the Sungai Lebir in the wet season (January 2020). For organism metabolism reduction and their activities in water, all collected water samples were kept at cool room temperature below 4 °C. Grab samples were collected and preserved in an icebox before being transported to the laboratory for chemical analysis. The sampling was carried out for one day. The weather was fine during sampling. On-site testing of pH, temperature, and dissolved oxygen (DO) was conducted. In the laboratory, physical and chemical parameters were measured, such as turbidity, biochemical oxygen demand (BOD), total suspended solids (TSS), ammonical nitrogen ($NH_3N$), iron (Fe), manganese (Mn), nickel (Ni), arsenic (As), mercury (Hg), and aluminum (Al). Chemical parameters were analyzed according to the standard methods [22]. The accuracy and precision of the data were above 95%. A summary of the data is given in Table 1.

**Table 1.** Summary of the data obtained from in-situ and laboratory measurements. ND is not detected.

| Parameters | Minimum | Maximum | Average | Standard Deviation |
|---|---|---|---|---|
| pH | 6.100 | 7.400 | 6.647 | 0.393 |
| Temperature (°C) | 27.300 | 30.000 | 28.805 | 0.570 |
| DO (mg/L) | 5.700 | 7.020 | 5.957 | 0.285 |
| BOD (mg/L) | 1.000 | 7.000 | 4.053 | 1.849 |
| TSS (mg/L) | 3.000 | 75.000 | 27.316 | 25.327 |
| $NH_3N$ (mg/L) | 0.010 | 2.200 | 0.598 | 0.755 |
| Fe (mg/L) | 0.005 | 2.136 | 0.770 | 0.761 |
| Turbidity (NTU) | 2.800 | 84.000 | 30.135 | 28.908 |
| Mn (mg/L) | 0.013 | 0.095 | 0.055 | 0.023 |
| Ni (mg/L) | ND | ND | ND | ND |
| As (mg/L) | ND | ND | ND | ND |
| Hg (mg/L) | ND | ND | ND | ND |
| Al (mg/L) | 0.091 | 1.657 | 0.686 | 0.544 |

A streamflow gauge station was situated at Sungai Lebir (approximately 58 km from upstream). The discharge data for the year 2020 were collected from the Department of Irrigation and Drainage, Malaysia. Channel slope was measured with a digital elevation model (30 m resolution) and Google Earth.

### 2.3. Methodology

2.3.1. Model Setup

A total of 63 reaches were considered for the model setup. Out of 63, 52 were on Sungai Lebir and 11 on Sungai Aring. Reach lengths were variable based on the tributaries and water treatment plant locations. Those tributaries originating from the mining site and joining either Sungai Lebir or Sungai Aring were considered point sources. Locations of diffused sources were along with the mining site, based on area slope (Figure 2). This assumption was based on the closeness of the mining area boundary with rivers, slope, and overflow direction.

Qual2k simulates discharge ($Q$) either in rectangular or trapezoidal-shaped channels. In this study, a trapezoidal-shaped channel was used. Channel side slopes ($s$) were estimated based on the river sinuosity. The channel bottom width ($B_0$) was estimated based on the top width, measured with Google Earth. Reach depth ($H$) was measured from in-situ

data [23]. Using side slopes, bottom width, and reach depth, the cross-sectional area ($A_c$) was calculated by [24]:

$$A_c = [B_o + 0.5(s_1 + s_2)H]H \tag{1}$$

Velocity ($v$) of reach was calculated by [19]:

$$v = \frac{Q}{A_c} \tag{2}$$

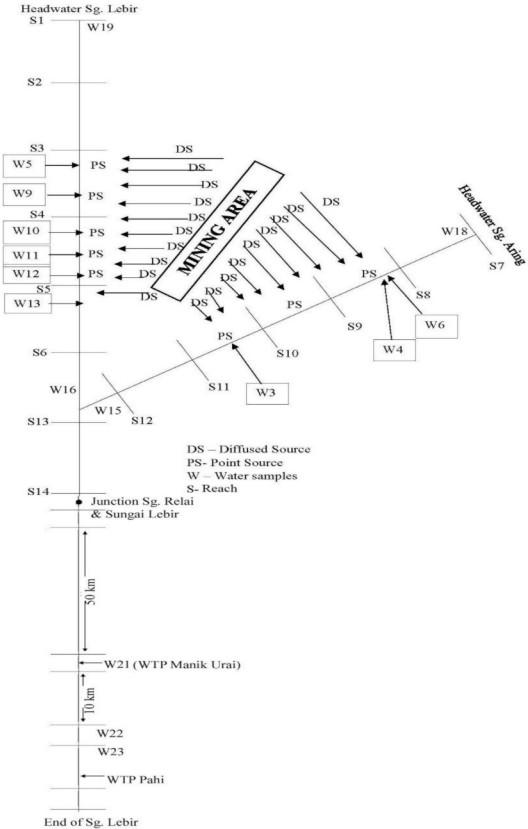

**Figure 2.** Schematic diagram of Qual2k modeling. Location of reaches, water samples, water treatment plants, diffused source, and non-diffused source is shown here.

Discharge for overland flow at the mining site was calculated using the widely used SCS curve number method [25]. The discharge was calculated for both wet and dry seasons to see their effect on pollution load. The overland flow discharge for wet and dry seasons was referred to as $Q_{r\_max}$ and $Q_{r\_min}$, respectively. The river discharge for the wet and dry seasons was referred to as $Q_{max}$ and $Q_{min}$, respectively. The general equation of the SCS curve number method is:

$$Q_{r\_max} \text{ or } Q_{r\_min} \left(\frac{mm}{h}\right) = \frac{(P_{r\_max} \text{ or } P_{r\_min} - I_a)^2}{(P_{r\_max} \text{ or } P_{r\_min} - I_a) + S} \tag{3}$$

$$I_a = 0.2 \times S \tag{4}$$

$$S = \frac{1000}{CN} - 10 \tag{5}$$

where $P$ (mm/h) is precipitation, $P_{r\_max}$ and $P_{r\_min}$ (mm/h) are precipitation in wet and dry seasons, respectively, $I_a$ is an initial abstraction, $S$ is potential maximum retention after runoff begins, and $CN$ is the curve number. According to the data (from 1990 to 2014) obtained from the Department of Irrigation and Drainage, Malaysia, $P_{r\_max}$ and $P_{r\_min}$ were

88 mm/h and 4 mm/h. Woolhiser [25] mentioned a table of curve numbers for different land uses. According to the study area's land use, the *CN* is 55.

### 2.3.2. Scenario Selection Criteria

According to the Department of Environment, Malaysia, water quality classes are classified from Class I (natural condition) to Class V (polluted, cannot be treated) [15]. Several studies compared water quality parameters with natural background levels to analyze pollution status [26,27]. In this study, Class I (natural condition) is considered the natural background level. Furthermore, the selected three scenarios are summarized in Table 2.

**Table 2.** A brief introduction of three scenarios used in the water quality modeling.

| Scenario | Assumptions |
|---|---|
| Baseline (Simulation of Natural Condition) | • Results from water samples were used as input for the baseline scenario, which shows the present natural condition in the study area.<br>• Values obtained from water samples were compared to National Water Quality Standards, Malaysia (NWQS) in order to estimate existing river quality. |
| Worst case (All mitigation measures failed) | • Assuming Urban Stormwater Management Manual for Malaysia (MSAM) guidelines, Land Disturbing Pollution Prevention and Mitigation Measures (LDP2M2) measures and other mitigation measures, such as detention ponds, failed to be implemented.<br>• Development activity plan failed to be followed.<br>• Water in tailing ponds failed and overflowed.<br>• River buffer is not in accordance with Department of Irrigation and Drainage (JPS) guidelines or inadequate.<br>• Removal of all vegetation from the mining site.<br>• Mining area is completely exposed to rainfall.<br>• Sediment discharge on all sides of the mining area during rainfall.<br>• Tributaries from the mining area to the main river carry extremely high total suspended solids (TSS).<br>• TSS values were assumed to be 10,000 mg/L which indicates that all mitigating measures failed during high-intensity rainfall in the study area. |
| With Mitigation (70% control of worst-case scenario) | • Mitigating measures are according to MSMA guidelines. Correct implementation of erosion and sediment control plan recommendations to control soil erosion from the site.<br>• Assuming site development activities are conducted in phases with LDP2M2. In this study, five silt traps are used. Each silt trap has an efficiency of 65% [28].<br>• River buffer is maintained in accordance with JPS guidelines.<br>• In addition to the above mitigation measures, additional measures such as detention ponds, phytoremediation application, chemical treatment, etc., are also implemented to those parameters which have high concentrations according to the NWQS. |

### 2.3.3. Pollution Load Calculation

The average pollution load for both wet and dry seasons was calculated by:

$$\text{Pollution Load} \left( \frac{kg}{day} \right) = Q_{r\_max} / Q_{r\_min} \left( \frac{m3}{s} \right) \times Cocncentration \left( \frac{mg}{L} \right) \times 86.4 \quad (6)$$

Calculated runoff for wet and dry season is summarized in Table 3.

**Table 3.** Calculation of runoff for wet and dry season.

| Parameters | High Flow | Low Flow |
|---|---|---|
| Rainfall, $P$ (mm/h) | 88 | 4 |
| Potential max Retention after Runoff begins, $S$ | 8.18 | 8.18 |
| Initial Abstractions, $I_a$ | 1.63 | 1.63 |
| Runoff, $Q$ (mm/h) | 78.89 | 0.53 |
| Area of mining site (m$^2$) | 2,024,000.0 | |
| $Q$ (m$^3$/s) | 44.35 | 0.29 |

2.3.4. Model Calibration and Validation

All parameters were calibrated and validated for both wet and dry seasons before qual2k modeling. The calibration was done by adjusting model coefficients and Manning's n values for each reach. Validation was performed by comparing observed and simulated data at a few locations due to the unavailability of data. The performance of calibration and validation between observed and simulated parameters was analyzed using the $R^2$ coefficient.

**3. Results**

*3.1. Current Condition of Physical Parameters (Baseline Scenario)*

A total of 3 physical parameters, such as temperature, total suspended solids (TSS), and turbidity, were observed at 23 locations near the mining site and along the reach. Slight variations were observed in temperature, which ranged from 27 to 30 °C with a standard deviation of 0.6 °C. Turbidity values varied, with a standard deviation of 28.9 NTU. High values (84 NTU) were observed upstream and at the river junction. Most turbidity values come under Class IIA according to the National Water Quality Standards of Malaysia (NWQS). This indicates that the water is slightly polluted and requires conventional treatment. Similarly, TSS values varied, with a standard deviation of 25.3 mg/L. High values (75 mg/L) were observed upstream and at the river junction that comes under Class III of NWQS. In terms of TSS, the water upstream and at the river junction are polluted but can be used for recreational activities with body contact.

*3.2. Current Condition of Chemical Parameters (Baseline Scenario)*

The pH along the reach ranged from 6.1 to 7.4 with a standard deviation of 0.4. This indicates that the water is suitable for drinking purposes. Dissolved oxygen (DO) is an important water quality parameter of a natural water system and a source of aquatic aerobic organisms. In the study area, DO varied from 5.7 mg/L to 7.02 mg/L with a standard deviation of 0.3 mg/L. It comes under Class IIA, which indicates the DO level is in natural condition. Biochemical oxygen demand (BOD) varied from 1 mg/L to 7 mg/L with a standard deviation of 1.8 mg/L. According to the NWQS, it is variable from Class I to Class III, which indicates polluted water at some locations. Ammonical nitrogen (NH$_3$N) varied from 0.01 mg/L to 2.2 mg/L with standard deviation of 0.75 mg/L. It was not detected at most of the locations up and midstream, which comes under Class I. However, high values were observed downstream, which comes under Class IV. This indicates that the downstream water is not suitable for drinking purposes. It is a great concern because two water treatment plants are located downstream.

*3.3. Current Condition of Heavy Metals (Baseline Scenario)*

Six heavy metals such as iron (Fe), manganese (Mn), nickel (Ni), arsenic (As), mercury (Hg), and aluminum (Al) were analyzed. Ni, As, and Hg were not detected along the reach. Fe varied from 0.005 to 2.1 mg/L with a standard deviation of 0.76 mg/L, which comes under natural conditions. High values (Class V) were observed at upstream and river junctions, which indicated high pollution and cannot be used either for drinking or irrigation purposes. Aluminum values ranged from 0.091 to 1.657 mg/L with a standard deviation of 0.5 mg/L, which comes under Class V (highly polluted). This indicates that high Al concentrations

were observed in existing conditions which makes the water not suitable for drinking or irrigation purposes. All values of Mn come under natural conditions.

### 3.4. Qual2k Model Calibration and Validation Results

During calibration of the wet season's flow velocity, Manning's n values varied from 0.055 to 0.600 s m$^{-1/3}$ with an average of 0.189 sm$^{-1/3}$. While, during calibration of the dry season's flow velocity, Manning's n values ranged from 0.85 to 4.9 s m$^{-1/3}$ with an average of 2.158 sm$^{-1/3}$. Before model calibration, the model underestimated parameter values in both wet and dry seasons. After calibration, the results showed that almost all parameters, including discharge and velocity, come under the excellent class ($R^2 > 0.85$) according to the model performance criteria [29]. This correlation indicates that observed parameters are comparable to simulated parameters. Model performance during validation was also accurate for all parameters. A summary of calibration and validation results during the wet and dry season are shown in Table 4.

**Table 4.** Summary of calibration and validation results. HF and LF are high and low flows, respectively.

| Parameters | Calibration ($R^2$ Values) | | | | Validation | | | |
| | Sg. Lebir | | Sg. Aring | | Sg. Lebir | | Sg. Aring | |
| | HF | LF | HF | LF | Observed | Simulated | Observed | Simulated |
|---|---|---|---|---|---|---|---|---|
| Discharge (m$^3$/s) | 0.98 | 0.99 | 0.98 | 0.99 | 1020.0 | 1021.4 | 240.0 | 240.0 |
| Velocity (m/s) | 0.97 | 0.96 | 0.96 | 0.96 | 0.499 | 0.492 | 0.692 | 0.672 |
| TSS (mg/L) | 0.97 | 0.96 | 0.98 | 0.98 | 34.00 | 35.35 | 67.00 | 65.88 |
| DO (mg/L) | 0.87 | 0.83 | 0.96 | 0.94 | 6.11 | 6.31 | 5.80 | 5.81 |
| BOD (mg/L) | 0.99 | 0.99 | 0.99 | 0.99 | 3.00 | 3.17 | 7.00 | 6.75 |
| NH3N (mg/L) | 0.99 | 0.99 | 0.99 | 0.99 | 1.020 | 1.057 | 0.010 | 0.010 |
| Fe (mg/L) | 0.95 | 0.95 | 0.99 | 0.99 | 0.770 | 0.745 | 1.909 | 1.880 |
| Mn (mg/L) | 0.92 | 0.91 | 0.99 | 0.98 | 0.049 | 0.050 | 0.042 | 0.040 |
| Al (mg/L) | 0.93 | 0.94 | 0.93 | 0.93 | 1.657 | 1.641 | 0.963 | 0.973 |
| pH | 0.97 | 0.97 | 0.96 | 0.96 | 6.90 | 7.08 | 6.10 | 5.88 |

### 3.5. Pollution Load of All Parameters

The calculated values of potential maximum retention after runoff begins ($S$) and $I_a$ were 8.18 and 1.64, respectively. $Q_{r\_max}$ and $Q_{r\_min}$ were calculated using Equations (3)–(5) as 44.35 m$^3$/s and 0.29 m$^3$/s, respectively. $Q_{r\_max}$ and $Q_{r\_min}$ in Equation (3) are mentioned in mm/h. It was converted into m$^3$/s with the help of mining area (m$^2$), conversion of "mm" to "m", and conversion of an hour to seconds. Based on 10-year daily rainfall data, the average daily rainfall distribution in the study area is 0 to 20 mm/h (98.7%), 20 to 40 mm/h (1.16%), and >40 (0.09%). $P_{r\_max}$ (worst case scenario) lies in the category > 40 mm/h. It means the possibility of a worst-case condition would be 0.1% or almost 36 days in a year. The average pollution load (kg/day) for the wet and dry season are given in Table 5.

**Table 5.** Pollution load in kg/day.

| Parameters | Pollution Loads | |
| | High Flow | Low Flow |
|---|---|---|
| DO | 17.28 | 17.28 |
| BOD | 14,369.4 | 96.48 |
| TSS | 864,000 | 1151.41 |
| NH3N | 70.25 | 0.47 |
| Iron | 2245.45 | 15.07 |
| Manganese | 201.49 | 1.35 |
| Nickel | 26.82 | 0.18 |
| Arsenic | 30.65 | 0.20 |
| Mercury | 19.15 | 0.12 |
| Aluminum | 2501.87 | 16.79 |

### 3.6. Worst-Case and with Mitigation Scenario

3.6.1. Total Suspended Solid (TSS)

The mining site is situated approximately 8 km upstream of Sungai Lebir and Sungai Aring. During the wet season, river flow is generally high (about 600 m³/s). As per the calculation of pollution load, it was considered that the minimum worst-case condition of TSS could be 10,000 mg/L. Upstream of Sungai Lebir, TSS concentration was 67.8 mg/L, which increased to 731 mg/L near the mining site. Similarly, at Sungai Aring, the concentration increased from 61.9 mg/L to 1591 mg/L near the mining site. At the Sungai Aring downstream, the concentration was 3507.2 mg/L.

At the junction of Sungai Lebir and Sungai Aring, the concentration increased from 1815.4 mg/L to 2274.6 mg/L. It increased continuously up to a maximum of 2698 mg/L at 32 km from Sungai Lebir upstream. After 32 km, the concentration started declining. At the water treatment plant (WTP) Manik Urai and Pahi, which are 68 km and 84 km from Sungai Lebir upstream, respectively, TSS concentrations were reported 2495 mg/L and 1482 mg/L. Downstream, the concertation was 1232 mg/L. It was observed that from 32 km to 62 km, the concentration declined with a standard deviation of 0.98 mg/. Whereas, from 62 km to downstream, the concentration declined with a standard deviation of 50.8 mg/L. The reason for the sudden change in standard deviation is a combination of river widening, less polluted water addition through tributaries, and precipitated water addition. However, no significant change in TSS concentration was observed in Sungai Lebir and Sungai Aring during the dry season.

In the "with mitigation" scenario, it was assumed that 70% could be controlled during the worst-case scenario. Also, five detention ponds were used. Each detention pond has 65% efficiency [28]. Results showed no significant changes compared to the baseline scenario if proper mitigation measures were applied. Variations in TSS concentration along the reaches are shown in Figure 3.

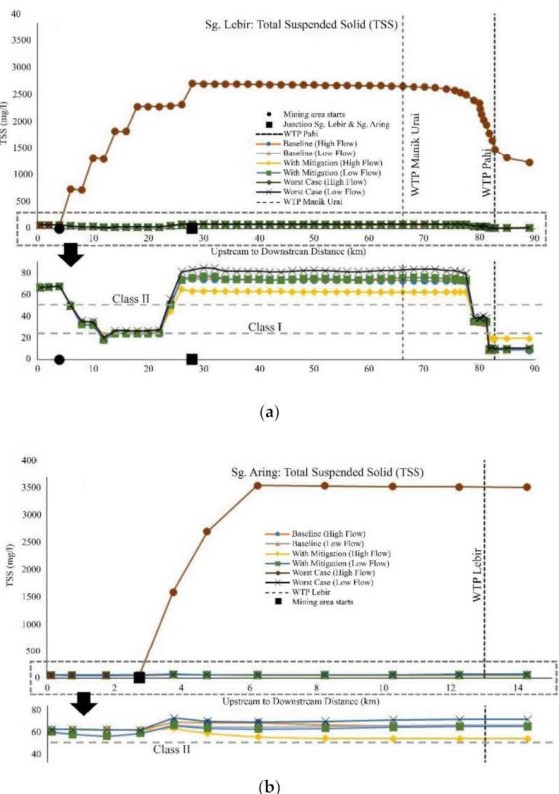

**Figure 3.** (**a**,**b**) showing TSS concentrations in different scenarios for Sungai Lebir and Sungai Aring, respectively.

### 3.6.2. Dissolved Oxygen (DO)

DO concentration declined from 5.8 mg/L (upstream) to 5.4 mg/L (near mining site). Whereas, at Sungai Aring, it declined from 5.96 mg/L (upstream) to 5.93 mg/L (near the mining site). The concentration was 2.45 mg/L at the junction of both reaches. The minimum concentration was observed at 76 km (0.86 mg/L) between WTP Manik Urai and Pahi. It showed that both WTPs were affected during the worst-case scenario. The concentration declination was almost constant with a standard deviation of 0.14 mg/L up to 76 km. At Sungai Aring, the standard deviation was 0.38 mg/L. The recovery starts after 76 km with a standard deviation of 0.07 mg/L. However, no significant change in DO concentration was observed in Sungai Lebir and Sungai Aring during the dry season.

In mitigation, the phytoremediation effect is also considered to improve DO concentration [30,31]. Results showed improvement in concentration compared to baseline if proper application of mitigation measures during the worst-case scenario and mining operations are applied. Variations in DO concentration along the reaches are shown in Figure 4.

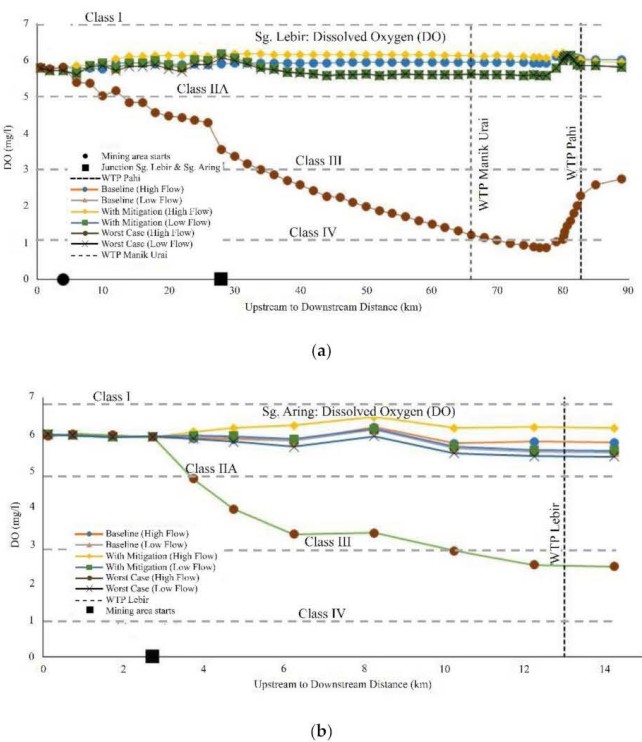

(a)

(b)

**Figure 4.** (**a**,**b**) showing DO concentration in different scenarios for Sungai Lebir and Sungai Aring, respectively.

### 3.6.3. Biochemical Oxygen Demand (BOD)

BOD concentration increased from 6.97 mg/L (Sg. Lebir upstream) to 13.95 mg/L (near the mining site). At the end of the mining site (11 km from the upstream), the concentration was 17.94 mg/L. Whereas, at Sungai Aring, it was increased from 3.96 mg/L (upstream) to 42.23 mg/L (near the mining site and 96.15 mg/L at the end of the mining site. After the reach junction, it increased from 24.17 mg/L to 38.89 mg/L. The maximum concentration was observed at 40 km (55.94 mg/L) and then gradually declined up to 68 km with a standard deviation of 0.03 mg/L. After 68 km, the declination rate was variable with a standard deviation of 0.94 mg/L. This variation is due to river widening and its relation with TSS concentration [32]. However, no significant difference was observed in BOD concentration during the dry season compared to the baseline scenario.

In the with mitigation scenario, by including the phytoremediation effect, results showed slightly lower concentration compared to baseline values. Variations in BOD concentration along the reaches are shown in Figure 5.

### 3.6.4. Ammonical Nitrogen (NH$_3$N)

NH$_3$N concentration increased from 0.01 mg/L (Sg. Lebir upstream) to 0.1 mg/L (near mining site). Compared to TSS, DO, and BOD, the sudden rise of NH$_3$N concentration was observed after the reach junction. Because of NH$_3$N concentration at Sg. Aring downstream was high (33.8 mg/L) due to the high pollution load. At Sg. Aring, NH$_3$N concentration increased from 0.01 mg/L (upstream) to 15.0 mg/L (near mining site). The concentration along the mining site increased with a standard deviation of 3.03 mg/L. After the junction, the concentration became 11.21 mg/L. Up to 76 km, the concentration increased with a standard deviation of 0.14 mg/L. After 76 km, the concentration declined with a standard deviation of 1.95 mg/L. At the Sg. Lebir downstream, the concentration was 5.82 mg/L which is higher compared to the baseline value (1.36 mg/L). No significant differences were observed in NH$_3$N concentration during the dry season compared to baseline values.

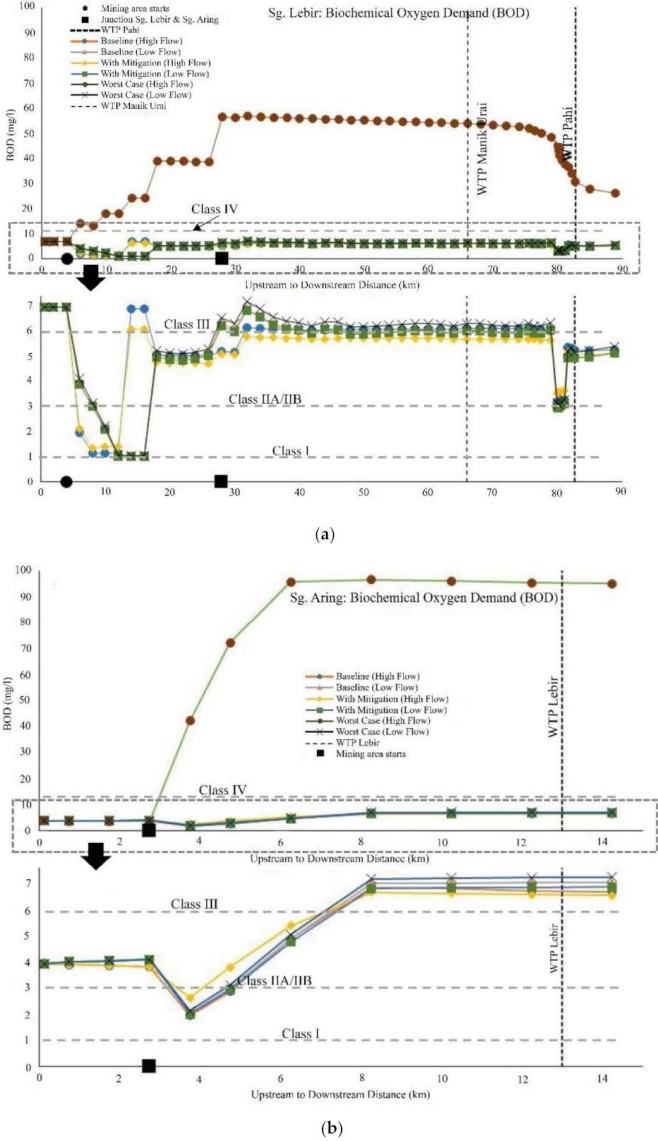

**Figure 5.** (**a**,**b**) showing BOD concentrations in different scenarios for Sungai Lebir and Sungai Aring, respectively.

In the with mitigation scenario, the concentration was still high in both reaches, which affects WTPs and cannot be used for drinking and irrigation purposes according to the National Water Quality Standards, Malaysia (NWQS). Variations in NH$_3$N concentration along the reaches are shown in Figure 6.

### 3.6.5. Iron (Fe)

At Sg. Lebir, Fe concentration increased from 2.06 mg/L (upstream) to 4.15 mg/L (near mining site). After 4 km, the concentration raised to 7.58 mg/L, which is almost the maximum in the worst-case scenario. It declined up to 26 km (river junction) with a standard deviation of 0.16 mg/L. After the junction, the concentration declined from 7.67 mg/L to 5.36 mg/L. Up to 79 km, it declined with a standard deviation of 0.15 mg/L. Further, it declined with a standard deviation of 0.74 mg/L up to the downstream. Whereas, at Sg. Aring, the concentration raised near the mining site from 1.68 mg/L to 10.68 mg/L and was almost constant up to the river junction. During the dry season, a slight increment was observed at both Sg. Lebir and Sg. Aring.

In the with mitigation scenario, no significant differences were observed. However, the Fe concentration is still high due to high baseline values according to NWQS. Variations in Fe concentration along the reaches are shown in Figure 7.

### 3.6.6. Aluminum (Al)

Al concentration at Sg. Lebir gradually increased from the mining site to the junction with a standard deviation of 1.32 mg/L. Whereas, at Sg. Aring, the concentration suddenly raised near the mining site and increased towards the junction with a standard deviation of 4.82 mg/L. Due to high concentration at Sg. Aring, the concentration after the junction increased from 5.08 mg/L to 11.24 mg/L. It started decreasing after 60 km with a standard deviation of 0.11 mg/L. However, no significant difference was observed during the dry season compared to the baseline scenario. Variations in Fe concentration along the reaches are shown in Figure 8.

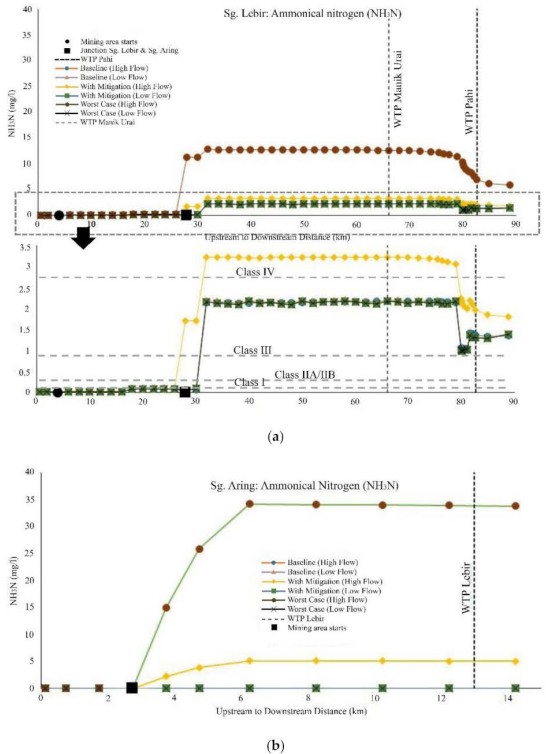

**Figure 6.** (**a**,**b**) showing NH$_3$N concentrations in different scenarios for Sungai Lebir and Sungai Aring, respectively.

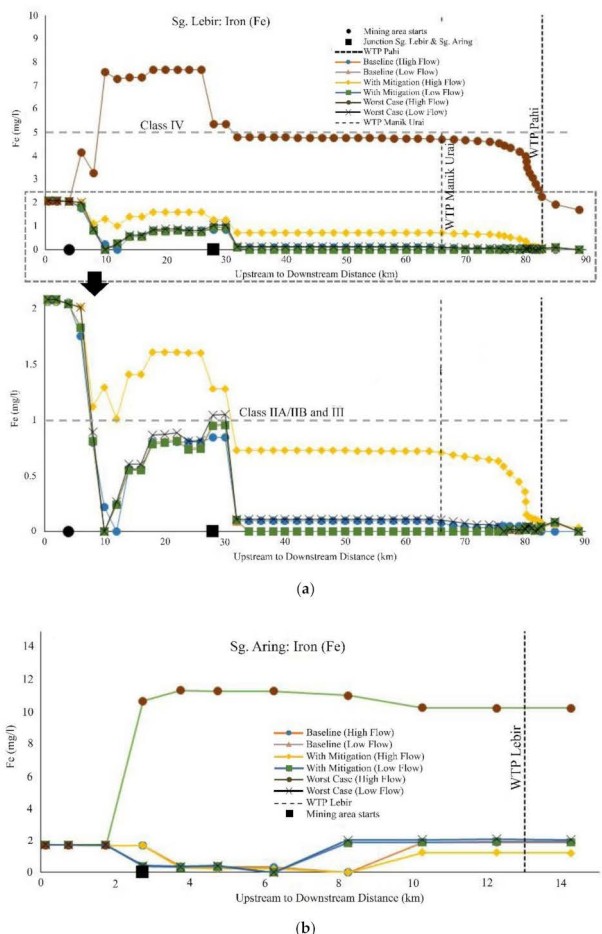

(a)

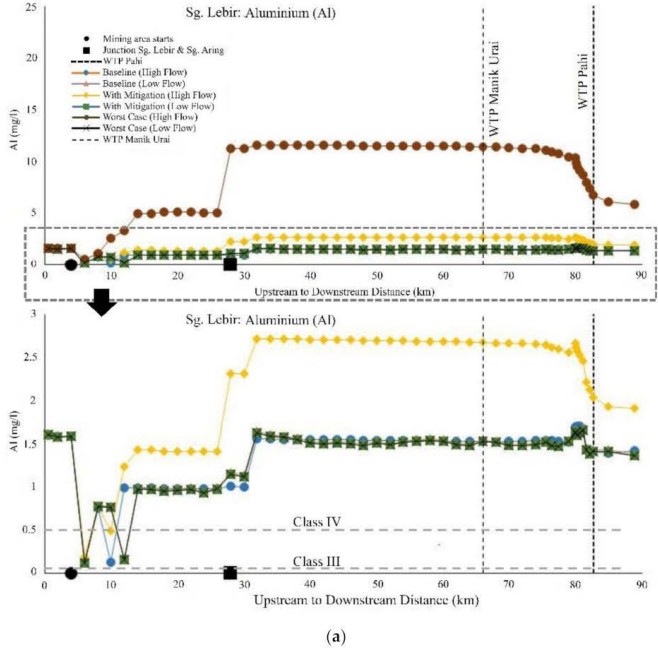

(b)

**Figure 7.** (**a**,**b**) showing Fe concentration in different scenarios for Sungai Lebir and Sungai Aring, respectively.

**Figure 8.** *Cont.*

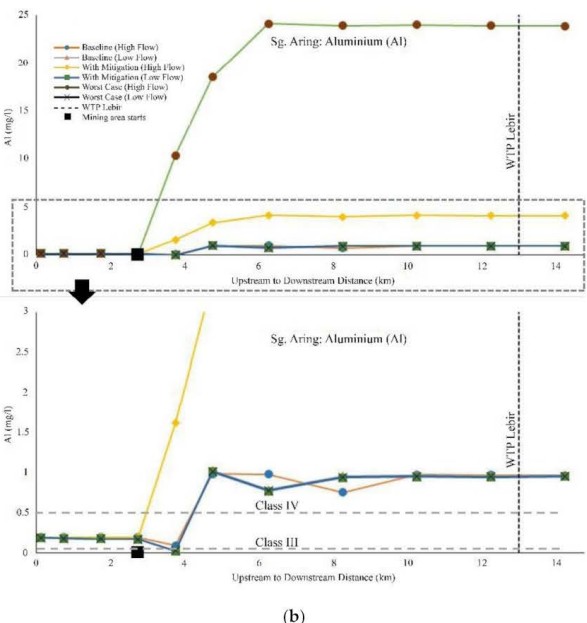

(b)

**Figure 8.** (**a**,**b**) showing Fe concentration in different scenarios for Sungai Lebir and Sungai Aring, respectively.

In the with mitigation scenario, high values compared to the baseline scenario were observed during the wet season. After the junction, the concentration increased from the baseline value (1.00 mg/L) to mitigation (2.31 mg/L). Similarly, at Sg. Aring, the concentration downstream increased from a baseline value (0.96 mg/L) to with mitigation value (4.11 mg/L).

### 3.6.7. Magnesium (Mg), Nickel (Ni), and Arsenic (As)

Mg, Ni, and As were not detected during data collection. Their values were under the natural condition in all scenarios except the wet season of a worst-case scenario, where a slight increment was observed. However, no significant differences were observed during the with mitigation scenario.

Average results of all parameters during the worst-case and with mitigation scenario are shown in Table 6.

**Table 6.** Average results of all parameters (mg/L) in worst-case and with mitigation scenarios.

| Parameters | Worst-Case Scenario | | | | With Mitigation Scenario | | | |
| --- | --- | --- | --- | --- | --- | --- | --- | --- |
| | Sg. Lebir High Flow | Sg. Aring Low Flow | Sg. Lebir High Flow | Sg. Aring Low Flow | Sg. Lebir High Flow | Sg. Aring Low Flow | Sg. Lebir High Flow | Sg. Aring Low Flow |
| TSS | 1337.9 | 52.9 | 1337.9 | 52.9 | 50.1 | 50.1 | 50.1 | 50.1 |
| DO | 3.2 | 5.7 | 3.2 | 5.7 | 5.96 | 5.72 | 5.96 | 5.72 |
| BOD | 29.4 | 6.1 | 29.4 | 6.1 | 5.9 | 5.9 | 5.9 | 5.9 |
| NH$_3$N | 6.1 | 1.1 | 6.1 | 1.1 | 1.7 | 1.1 | 1.7 | 1.1 |
| Fe | 2.8 | 0.7 | 2.8 | 0.7 | 0.91 | 0.69 | 0.91 | 0.69 |
| Mn | 0.22 | 0.03 | 0.22 | 0.03 | 0.061 | 0.035 | 0.061 | 0.035 |
| Ni | 0.055 | 0.006 | 0.055 | 0.006 | 0.013 | 0.006 | 0.013 | 0.006 |
| As | 0.052 | 0.0078 | 0.052 | 0.0078 | 0.013 | 0.007 | 0.013 | 0.007 |
| Hg | 0.033 | 0.004 | 0.033 | 0.004 | 0.008 | 0.005 | 0.008 | 0.005 |
| Al | 6.361 | 1.490 | 6.361 | 1.490 | 2.1 | 1.4 | 2.1 | 1.4 |

## 4. Discussion

Water quality parameters were analyzed and compared based on National Water Quality Standards, Malaysia (NWQS). According to the NWQS, temperature, TSS, and

turbidity are either in a natural condition or slightly polluted, which requires conventional treatment. In terms of these parameters, three WTPs come under Class I, which indicates natural conditions. Baseline DO concentration is slightly high from upstream to downstream, which comes under Class IIA. Baseline BOD concentration is also variable from upstream to downstream of the reach. DO and BOD's high baseline values are due to the upstream location of the study area, where there are several anthropogenic activities such as ore and palm oil factories, logging, and sand mining activities along rivers. These activities cause soil erosion and weathering of sedimentary rock containing heavy metals, which flow into nearby rivers [33]. NH$_3$N values are high at two WTPs (near downstream) due to the joining of the stream (Sungai Sok) passing through settlement areas. High NH$_3$N concentration may be caused by waste released into the stream from factories, residential, and agricultural areas situated along the stream [34].

Fe concentrations were high upstream of both Sg. Lebir and Sg. Aring. However, these values are lower near the mining site and WTPs. In the baseline condition, no Fe effect was observed. Baseline values of Al are high from upstream to downstream of the reach, which is a great concern. In the current situation, all WTPs required chemical treatment to reduce the Al effect. High concentrations of both Fe and Al are due to the weathering and erosion process of igneous and sedimentary rocks through anthropogenic activities [33]. Also, during mining, all significant mitigation measures must be applied to lower the Al effect on the reach.

During the worst-case scenario of the wet season, TSS concentration comes under Class V, which indicates that the water cannot use for drinking and irrigation purposes. During the worst-case scenario of the dry season, the concentration comes under Class II. This indicates that conventional treatment is required and sensitive to aquatic species. In the with mitigation scenario in both seasons, the concentration is similar to baseline values, which indicates suggestive preventive measures are better for controlling TSS concentration during mining.

DO concentration, during the worst-case scenario of the wet season, comes under Class V from the mining site to downstream. This indicates that the water cannot be used for drinking and irrigation purposes. Based on the with mitigation scenario, the concentration improved compared to baseline values. It showed that water quality would not be affected if proper mitigation measures were applied. Similar results were found for BOD concentration. NH$_3$N concentration comes under Class V after the junction at Sg. Lebir and near the mining site at Sg. Aring. This showed that NH$_3$N concentrations were not much affected in the worst-case scenario of the wet season near the mining site. Furthermore, the role of phytoremediation and denitrification played an important role in controlling BOD and NH$_3$N concentration, respectively, in the worst-case scenario.

Fe and Al concentrations come under Class V during the worst case of the wet season. A high concentration was observed near the mining site and continued downstream. This indicates that WTPs were affected due to the high concentration. The concentrations were similar to baseline values during the with mitigation scenario. This also shows that WTPs will not be affected if proper mitigation measures are applied.

Overall, worst-case scenario condition chances would be 0.1% (or 36 days) based on 10-year daily precipitation data. These days, more focus will be required on mitigation measures to control the worst-case scenario.

## 5. Conclusions

Water quality parameters, such as temperature, TSS, turbidity, DO, and BOD, were slightly polluted in baseline or existing condition due to anthropogenic activities. NH$_3$N, Mg, Ni, As, and Hg were not detected at all sampling points, except NH$_3$N at a few locations, due to industrial and agricultural waste. Fe at most places was in natural condition except at upstream and river junctions due to weathering and erosion. Whereas Al concentration was high at all places, which could be due to waste discharge at several locations along the reach and other mining activities situated upstream of Sg. Lebir and

Sg. Aring. During the worst scenario of the wet season, almost all parameters were at a high concentration, which starts from the near mining site and continues up to Sg. Lebir downstream. However, the variation pattern of all parameters was different depending on pollution load, baseline values, and channel velocity. WTPs during this condition were affected due to high parameter concentration. During the worst case of the dry season, no significant difference was observed compared to baseline values, which indicates very low chances of pollution. Results also showed that if proper mitigation measures are applied, such as phytoremediation, denitrification, detention ponds, and chemical treatment of heavy metals, the worst-case impact can significantly be reduced. It also reduces the impact on WTPs, which is one of the main concerns in environmental impact assessment.

The study also revealed that the selection and analysis of three scenarios were helpful in the critical environmental impact assessment of mining activities. This study will also be significant to government and private agencies for environmental impact assessment clearance and protection of WTPs, irrigation water supply, and aquatic species. Furthermore, this study can be improved by collecting detailed in-situ data such as discharge measurement at several locations, parameters measurement in different seasons, and increment in modeling time.

**Author Contributions:** Conceptualization, M.T.A., M.O.A.K., N.K.B.E.M.Y., Z.J. and F.M.O.; methodology, M.T.A. and M.O.A.K.; software, M.T.A.; validation, M.T.A.; formal analysis, M.T.A. and M.O.A.K.; investigation, M.T.A., M.O.A.K. and F.M.O.; resources, M.O.A.K., F.M.O., N.K.B.E.M.Y. and Z.J.; data curation, M.O.A.K. and F.M.O.; writing—original draft preparation, M.T.A.; writing—review and editing, M.T.A., M.M.A.K., K.A., A.M.E., P.A., M.S.F., M.O.A.K. and F.M.O.; visualization, M.T.A., M.M.A.K., M.O.A.K., A.M.E., P.A., N.K.B.E.M.Y., Z.J., M.S.F. and F.M.O.; supervision, M.M.A.K. and M.O.A.K.; project administration, M.M.A.K., M.O.A.K. and F.M.O.; funding acquisition, K.A., A.M.E., P.A. and M.S.F. All authors have read and agreed to the published version of the manuscript.

**Funding:** This research was supported by Researchers Supporting Project number (RSP-2021/249), King Saud University, Riyadh, Saudi Arabia.

**Acknowledgments:** We gratefully acknowledge the Universiti Sains Malaysia and Pultex Environment Sdn. Bhd. for providing the required research facilities and data for this work. We are also thankful to the Universiti Kelantan Malaysia for providing additional support in this study. We would like to thank King Saud University for financial support by Researchers Supporting Project number (RSP- 2021/249), King Saud University, Riyadh, Saudi Arabia.

**Conflicts of Interest:** The authors declare no conflict of interest.

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
