# Peer review of "A New Scenario-Based Approach for Water Quality and Environmental Impact Assessment Due to Mining Activities"

_water, doi:10.3390/w14132117_

Round 1

Reviewer 1 Report

The manuscript titled “A new scenario-based approach for water quality and environmental impact assessment due to mining activities” aims to study water quality in areas of mining activities in Malaysia. The study was conducted at Sungai (river) Lebir and Sungai Aring situated in Gua Musang, Kelantan, Peninsular Malaysia with the objective to assess spatio-temporal environmental impact due to mining activities in wet and dry seasons.

Minor grammatical correction, Partial collection included with highlighted sentences. The authors are warmly encouraged to restructure and revise some sentences.

In the introduction, the authors should start by identifying knowledge gaps of broad international interest which can be of methodological nature, associated with process understanding, or rotating around relevant and innovative empirical evidence. It is fundamental, though, that the innovative aspects and elements are of general and international interest and useful to other scientists, especially water quality is of interest to all of us.

Methods seems OK, however, quantification of water quality needs to be highlighted. Perhaps a table of water quality by quantifying contents will be very useful.

A high resolution graphics of figure 1 will also improve the manuscript.

Reviewer 2 Report

This study assessed spatio-temporal environmental impact due to mining activities in wet and dry seasons. Several parameters data were collected at different locations along the reach. Point and non-point sources were considered near mining site. Overland flow at the mining site was calculated by widely used SCS (Soil Conservation Service) curve number method. Furthermore, this study will be helpful in pre and post mining assessment and for environ-mental impact clearance. This manuscript contains interesting information. However, there are some weaknesses, a major revision is needed before publish. Specific comments are as follow:

1. The introduction does not summarize the research progress well and should be rewritten it, which must contain the research progress of the environmental impact due to mining activities, evaluation method, the environmental problems and pollutants of the study area. In addition, Part of the references are too old.

2. It is necessary to further highlight the new method and clarify its characteristics.

3. Why only choose nutrients and heavy metals?

4. Most analyses are simple descriptions of the data. Please further mining the rules of data and increasing comparative analysis.

5. The conclusion needs further elaboration.

Reviewer 3 Report

The purpose of the manuscript “water_ 1790817” are to conduct water quality modeling along the reaches using Qual2k, to  examine the spatio-temporal variation of parameter concentration during the wet and dry  season, and to analyze the pollution effect on water treatment plants during the wet  and dry season.

The paper appears well-structured; however, some sections must be improved.  Therefore, I believe

the manuscript should be published only after major revision.

Comments (R = row#):

R=44: Geogenic pollution is few o never mentioned in the introduction, which should still be taken into consideration. I recommend reading the following works and integrating this part:

Liu, Y., Xiao, T., Zhu, Z., Ma, L., Li, H., & Ning, Z. (2021). Geogenic pollution, fractionation and potential risks of Cd and Zn in soils from a mountainous region underlain by black shale. Science of the Total Environment, 760, 143426.

Fuoco I., De Rosa R., Barca D., Figoli A., Gabriele B. and Apollaro C., (2022). Arsenic polluted waters: Application of geochemical modelling as a tool to understand the release and fate of the pollutant in crystalline aquifers. Journal of Environmental Management, 301, p.113796.

DOI:10.1016/j.jenvman.2021.113796

Fuoco, I., et al. "Use of reaction path modelling to investigate the evolution of water chemistry in shallow to deep crystalline aquifers with a special focus on fluoride." Science of The Total Environment 830 (2022): 154566.

R:98: A geological description of the site is completely absent. A minimum of geological description and a schematic geological map should be included

R:130: Some parameters should be measured in situ, such as pH and DO, it makes little sense to determine them in the laboratory ..

the data of the precision and analytical accuracy are missing and a table with the data should be inserted in the work

R:216: it would be necessary to specify well the natural origin of some elements, also referring to the natural background values in at least similar contexts. See for example the works of:

Apollaro, C., Di Curzio, D., Fuoco, I., Buccianti, A., Dinelli, E., Vespasiano, G., Castrignanò, A., Rusi, S., Barca, D., Figoli, A. and Gabriele, B., 2022. A multivariate non-parametric approach for estimating probability of exceeding the local natural background level of arsenic in the aquifers of Calabria region (Southern Italy). Science of the Total Environment, 806, p.150345.

Lim, D. I., Choi, J. Y., Jung, H. S., Choi, H. W., & Kim, Y. O. (2007). Natural background level analysis of heavy metal concentration in Korean coastal sediments. Ocean and Polar Research, 29(4), 379-389.

for As

R:226: Are the authors sure that the elements they speak of refer to the solution and are not instead suspended material? has the sample been filtered?

The writings in all the figures are badly read

Discussions and conclusions need to be rewritten taking into account previous comments

Recommended works must be added in the bibliography

Liu, Y., Xiao, T., Zhu, Z., Ma, L., Li, H., & Ning, Z. (2021). Geogenic pollution, fractionation and potential risks of Cd and Zn in soils from a mountainous region underlain by black shale. Science of the Total Environment, 760, 143426.

Fuoco I., De Rosa R., Barca D., Figoli A., Gabriele B. and Apollaro C., (2022). Arsenic polluted waters: Application of geochemical modelling as a tool to understand the release and fate of the pollutant in crystalline aquifers. Journal of Environmental Management, 301, p.113796.

DOI:10.1016/j.jenvman.2021.113796

Fuoco, I., et al. "Use of reaction path modelling to investigate the evolution of water chemistry in shallow to deep crystalline aquifers with a special focus on fluoride." Science of The Total Environment 830 (2022): 154566.

Apollaro, C., Di Curzio, D., Fuoco, I., Buccianti, A., Dinelli, E., Vespasiano, G., Castrignanò, A., Rusi, S., Barca, D., Figoli, A. and Gabriele, B., 2022. A multivariate non-parametric approach for estimating probability of exceeding the local natural background level of arsenic in the aquifers of Calabria region (Southern Italy). Science of the Total Environment, 806, p.150345.

Lim, D. I., Choi, J. Y., Jung, H. S., Choi, H. W., & Kim, Y. O. (2007). Natural background level analysis of heavy metal concentration in Korean coastal sediments. Ocean and Polar Research, 29(4), 379-389.

Round 2

Reviewer 2 Report

Questions were answered and the articles have been improved 

Reviewer 3 Report

Remarks from reviewers have been correctly addressed, and the paper is now more focuse on his core topic
In my opinion it is now acceptable.
Best regards